# Tracking the Origin and Evolution of Diagenetic Fluids of Upper Jurassic Carbonate Rocks in the Zagros Thrust Fold Belt, NE-Iraq

**Namam Salih [1,2,\*], Alain Préat [3], Axel Gerdes [4], Kurt Konhauser [5] and Jean-Noël Proust [6]**

1 Engineering Faculty, Petroleum Engineering Department, Soran University, Soran-Erbil 44008, Iraq
2 Scientific Research Centre (SRC), Soran University, Soran-Erbil 44008, Iraq
3 Res. Grp.—Biogeochemistry & Modelling of the Earth System, Université Libre de Bruxelles, 1050 Brussels, Belgium; apreat@ulb.be
4 Institute for Geosciences, Goethe-University Frankfurt, Altenhöferallee 1, 60438 Frankfurt am Main, Germany; gerdes@em.uni-frankfurt.de
5 Department of Earth and Atmospheric Sciences, University of Alberta, Edmonton, Alberta AB T6G 2E3, Canada; kurtk@ualberta.ca
6 University Rennes, CNRS, Geosciences Rennes, UMR6118 35000 Rennes, France; proust@univ-rennes1.fr
\* Correspondence: namam.salih@soran.edu.iq

**Abstract:** Utilizing sophisticated tools in carbonate rocks is crucial to interpreting the origin and evolution of diagenetic fluids from the Upper Jurassic carbonate rocks along the Zagros thrust-fold Belt. The origin and evolution of the paleofluids utilizing in-situ strontium isotope ratios by high resolution laser ablation ICP-MS, integrated with stable isotopes, petrography and fieldwork are constrained. Due to the lack of information on the origin of the chemistry of the fluids, the cements that filled the Jurassic carbonate rocks were analysed from the fractures and pores. This allowed us to trace the origin of fluids along a diagenetic sequence, which is defined at the beginning from the sediment deposition (pristine facies). Based on petrography and geochemistry (oxygen-, carbon- and strontium-isotope compositions) two major diagenetic stages involving the fluids were identified. The initial stage, characterized by negative $\delta^{13}C_{VPDB}$ values (reaching −10.67‰), involved evaporated seawater deposited with the sediments, mixed with the input of freshwater. The second stage involved a mixture of meteoric water and hot fluids that precipitated as late diagenetic cements. The late diagenetic cements have higher depleted O–C isotope compositions compared to seawater. The diagenetic cements display a positive covariance and were associated with extra-$\delta^{13}C_{VPDB}$ and $\delta^{18}O_{VPDB}$ values (−12.87‰ to −0.82‰ for $\delta^{18}O_{VPDB}$ and −11.66‰ to −1.40‰ for $\delta^{13}C_{VPDB}$ respectively). The distinction between seawater and the secondary fluids is also evident in the $^{87}Sr/^{86}Sr$ of the host limestone versus cements. The limestones have $^{87}Sr/^{86}Sr$ up to 0.72859, indicative of riverine input, while the cements have $^{87}Sr/^{86}Sr$ of (0.70772), indicative of hot fluid circulation interacting with meteoric water during late diagenesis.

**Keywords:** origin of diagenetic fluids, strontium isotope-laser ablation ICP-MS, upper jurassic carbonate rocks, ZTFB, NE-Iraq

## 1. Introduction

The Upper Jurassic Barsarin formation is located along the Zagros thrust-fold belt (ZTFB) in NE Iraq, and is considered a giant undiscovered Jurassic source rock [1]. Several studies have documented the main petrographic features of the Barsarin formation in order to characterize the paleoenvironment, but without any attention to diagenesis. The formation comprises laminated limestones and dolomitic limestones, in places cherty, with autoclastically brecciated beds admixed with shaly and marly materials [2]. The micro-crystalline quartz (chert) associated with carbonate rocks have different silica

sources, including a biogenic origin [3], silica-enriched seawater [4], and/or linked to silica input via a river source [5].

The origin and evolution of the diagenetic fluids that affected the Barsarin sediments are still largely unknown due to a lack of detailed studies. However, the fluids can be traced by geochemistry, and particularly by analyses of stable (carbon and oxygen) and radiogenic (strontium) isotopes. Oxygen isotope compositions can provide insights into the origin of mixed fluids or cross-formation water flows, and also to characterize paleo-temperatures and rainfall properties, such as the amount, seasonally, and moisture sources (e.g., [6–8]). Carbon-isotope compositions can provide information on effective rainfall and weathering processes [8]. The conventional $^{87}Sr/^{86}Sr$ isotope composition is classically used to infer the main sources of radiogenic strontium in sedimentary basins related to the pulses of mid-oceanic hydrothermal fluxes [9] or to infer overprinting by radiogenic dolomitizing fluids (e.g.,[10]). It allows for the determination of continental riverine input [11]. However, the classical conventional technique to measure radiogenic isotopes is limited to the achievable spatial resolution of $^{87}Sr/^{86}Sr$ records.

To address the time specific gap in tracking the marine limestone and associated diagenetic fluids in the Barsarin formation, a new approach is developed here. It consists of coupling a high-resolution laser ablation (LA-) MC-ICP-MS analysis with fieldwork data, petrography, and geochemistry (oxygen-carbon stable isotopes). The remarkable high precision of laser ablation analysis is utilized to measure, for the first time in our sediments, the absolute radiogenic composition of the Jurassic carbonate rocks of the Barsarin formation, and evaluate the precise origin, involvement and evolution of different fluids during diagenesis that modified the original composition of the host limestones. This will also give information on the dynamic of the diagenetic fluids during the Zagros orogeny within the ZTFB.

## 2. Geological Setting

The Upper Jurassic high-folded zone, considered as a part of the Zagros fold-thrust belt, belongs to a NE-SW basin (Figure 1), inherited from the tectonic activity of the Arabian plate and the opening of the southern Neo-Tethys ocean [12]. This ocean was characterized by sea level fluctuations that affected evaporitic sabkhas in the basin during Jurassic-Cretaceous times. NE Iraq formed an euxinic basin separated from the Neo-Tethys by a rifted area where shallow water carbonates formed [12]. It is generally assumed that the Barsarin formation records an euxinic environment, subjected to a low subsidence that characterized the Neo-Tethys during this period.

The Barsarin formation is composed of laminated limestones and dolomitic limestones, and in places brecciated texture and crumbled and contorted beds have been also documented in the type locality and type section. The author also recognized a mixture of shaly and marly materials with melikaria structures, and the upper and lower contact boundaries of the Barsarin formation are identified by the Chia Gara formation and Naokelekan formation, respectively.

The Barsarin formation was previously described as an isolated lagoonal environment, mostly evaporitic, and based on stratigraphic position from fieldworks, it has a Kimmeridgian-Tithonian age [13]. The recent studied section of the Barsarin formation is located close to the Rawandus area where the series is considered as the reference section in northeastern Iraq (Figure 1).

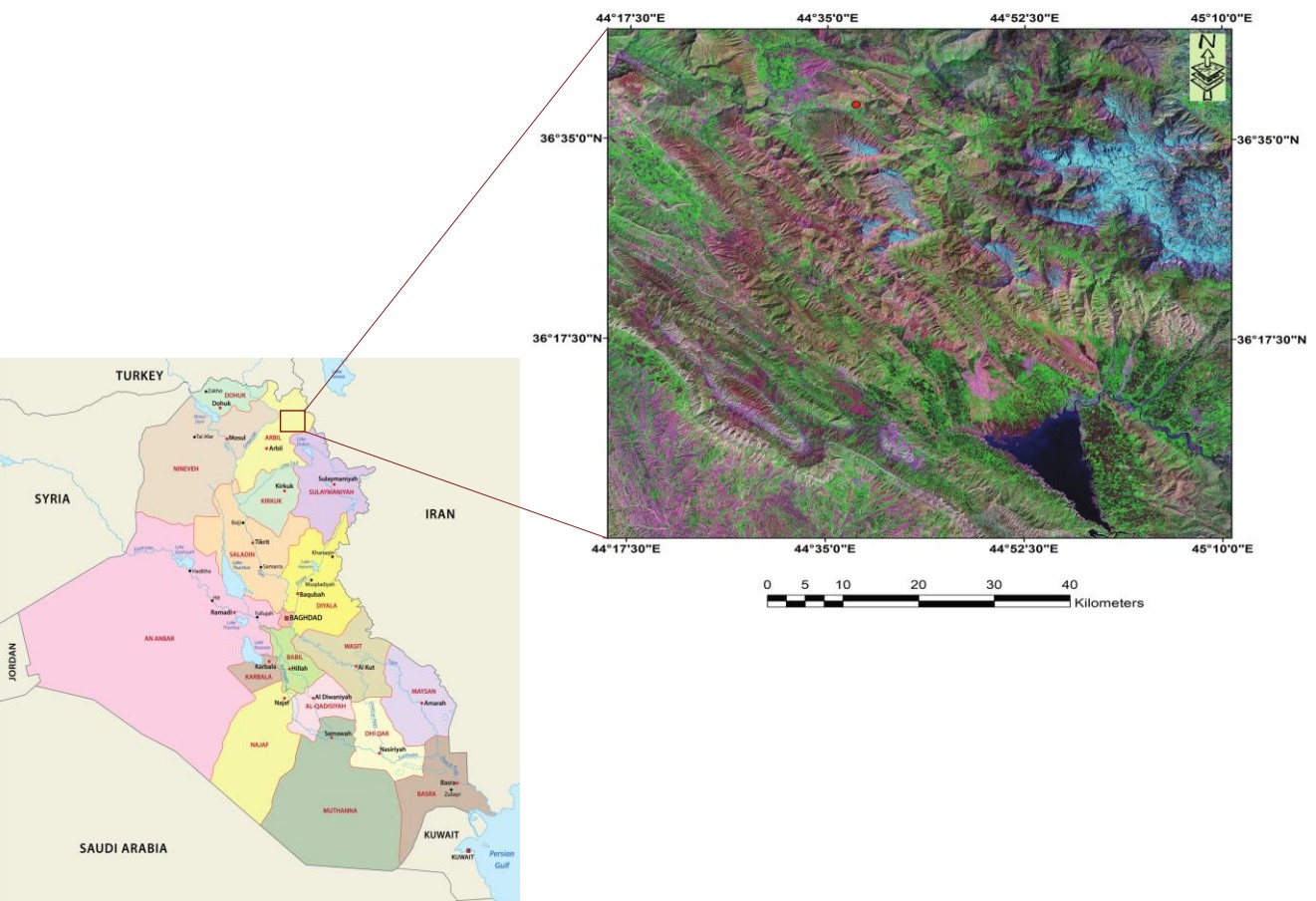

**Figure 1.** General map illustrates the location of the studied area "green circle".

## 3. Methods

Twenty-three samples were collected from the outcrop of the Barsarin formation. All the samples were prepared in laboratory for thin sections and studied under the optical microscope to distinguish the different carbonate phases, with a particular attention on the geometric cross-cut relations of fractures and veins. Scanning electron microscopy (SEM) was also used to image the surfaces to provide information about the morphology or texture of the surface. Selected samples of broken fragments (chips) mounted on aluminum stubs were studied from samples coated with gold depending on the purpose of work. SEM and EDX are also utilized in the recent study, backscattered image mode was used to give the different in contrast between minerals with different atomic number. The low atomic number samples give low emissions of backscattered electrons, while high atomic number samples give high emissions of these electrons. The backscattered electrons have higher energies than secondary electrons—usually from approximately 8 × 10-18 J (50 eV) up to the energy of the primary beam electrons.

The oxygen and carbon isotopic compositions of (24) samples were analyzed from powders, after selective microdrillings of the different recognized carbonate phases. However, the co-occurrence of fracture-filling calcite and dolomite, and also evaporites in the host limestones, made it difficult to avoid a mixing between these phases. As a consequence, the dolomite and calcite are considered as one group, and host limestone and evaporite as a second group.

Carbonate powders were reacted with 100% phosphoric acid at 70 °C using a Gasbench II connected to a Thermo Fisher Delta V Plus mass spectrometer. All values are reported in per mil relative to V-PDB (Table 1). Reproducibility and accuracy were monitored by replicate analysis of laboratory standards calibrated to international standards NBS19, NBS18 and LSVEC. Laboratory standards were calibrated by assigning

$\delta^{13}C_{VPDB}$ values of +1.95‰ to NBS19 and −46.6‰ to LSVEC and by assigning $\delta^{18}O_{VPDB}$ values of −2.20‰ to NBS19 and −23.2‰ to NBS18. The analyses (23) were performed in the University of Erlangen.(Germany, M. Joachimski).

**Table 1.** $\delta^{13}C$ (% VPDB), $\delta^{18}O$ (% VPDB) values of selected samples from Barsarin formation (n = 24).

| Sample No. | $\delta^{13}C_{VPDB}$ | $\delta^{18}O_{VPDB}$ |
|:---:|:---:|:---:|
| B.4 | −9.3 | −0.8 |
| B.15 | −7.5 | −2.9 |
| B.19 | −5.6 | −5.9 |
| B.5 | −9.6 | −3.5 |
| B.3 | −8.1 | −1.3 |
| B.11 | −8.2 | −3.8 |
| B.11 | −8.4 | −3.9 |
| B.12 | −7.0 | −1.4 |
| B.8 | −10.7 | −4.3 |
| B.0 | −8.7 | −0.3 |
| B.0 | −5.1 | −12.5 |
| B.6 | −4.9 | −12.9 |
| B.6 | −11.7 | −5.1 |
| B.22 | −8.2 | −1.2 |
| B.B | −8.6 | −9.6 |
| B.9 | −2.5 | −11.3 |
| B.20 | −6.3 | −1.7 |
| B.21 | −5.2 | −2.3 |
| B.14 | −7.5 | −0.8 |
| B.14 | −6.2 | −6.4 |
| B-Contact | −10.1 | −4.1 |
| B.16 | −7.1 | −2.3 |
| B.1 | 1.5 | −7.5 |
| B.10 | −11.0 | −4.9 |

Strontium isotope measurements on carbonate samples were performed by Laser Ablation ICP-MS at Goethe University-Frankfurt, using a Thermo-Finnigan Neptune multicollector (MC)-ICP-MS system attached to a Resolution 193 nm ArF Excimer laser ablation system (ComPexPro 102F, Coherent), equipped with an S-155 two-volume (Laurin Technic, Australia) ablation cell. Square laser spots with diameters or edge lengths of around 235 μm were drilled with an 8-Hz repetition rate, and energy density of about 6–7 J.cm$^{-2}$, during 45s of data acquisition.

The collector set-up included (1) $^{83}$Kr as a monitor to verify the successful correction for the isobaric interferences of $^{84}$Kr on $^{84}$Sr and of $^{86}$Kr on $^{86}$Sr by subtraction the gas blank measured before sample ablation; (2) mass 83.5 to gauge the production rate of doubly charged $^{167}$Er and calculate the production of doubly charged $^{164}$Er, $^{166}$Er, $^{168}$Er and $^{170}$Er, which interfere with $^{83}$Kr and $^{84}$Sr, $^{85}$Rb and $^{86}$Sr, respectively; (3) mass 86.5 to gauge the production rate of doubly charged $^{173}$Yb and calculate the production of doubly charged $^{168}$Yb, $^{172}$Yb, $^{174}$Yb and $^{176}$Yb, which interfere with $^{84}$Sr, $^{86}$Sr, $^{87}$Sr and $^{88}$Sr; (4) $^{86}$Sr and $^{88}$Sr to use $^{88}$Sr/$^{86}$Sr for mass bias correction; (5) $^{84}$Sr to check the accuracy of the mass bias and interference (see above) correction using $^{87}$Sr/$^{86}$Sr; (6) $^{85}$Rb to correct for the isobaric interference of $^{87}$Rb on $^{87}$Sr and to get the $^{87}$Rb/$^{86}$Sr; and (7) $^{87}$Sr to use in the radiogenic isotope ratio $^{87}$Sr/$^{86}$Sr.

At the beginning of the analytical session, a soda-lime glass SRM-NIST 610 was measured to three times for empirical determination of 87Rb/85Rb mass bias. The procedure yielded, after interference correction on $^{86}Sr$, $^{88}Sr$, and $^{85}Rb$ from doubly charged Yb and Er, the $^{87}Rb/^{86}Sr$ ratio needed for accurate correction of the isobaric interference of $^{87}Rb$ on $^{87}Sr$. An isotopically homogeneous in-house plagioclase standard (MIR-1) was measured throughout the analytical session to monitor accuracy and apply corrections to the measured unknowns, if necessary. This plagioclase is a megacryst from a lava of the Dutsin Miringa Hill volcano (Northern Cameroon Line; [14]) that has been independently characterized for $^{87}Sr/^{86}Sr$.

The reference materials BHVO-1 and BCR-2G were measured along with the in-house standard to monitor accuracy, in particular of the correction of the isobaric interference of $^{87}Rb$, and also the reproducibility of the results. The corrections for the presence of Rb were minor and the results for the unknowns, reported in Table 2, are considered to be accurate. Sr concentrations compared with MIR (3500 ppm), applying the same corrections as for MIR. All analyses together are given in Table 2.

**Table 2.** Strontium concentration values and $^{87}Sr/^{86}Sr$ ratios by laser spots analyses using ICP-MS (B9, B14, n = 35).

| Sample No. | Line No. | Sr ppm | $^{87}Sr/^{86}Sr$ |
|---|---|---|---|
| B9 | 109.00 | 148 | 0.70767 |
| B9 | 110.00 | 160 | 0.70772 |
| B9 | 111.00 | 191 | 0.70767 |
| B9 | 112.00 | 690 | 0.70750 |
| B9 | 113.00 | 496 | 0.70744 |
| B9 | 114.00 | 154 | 0.70768 |
| B9 | 115.00 | 594 | 0.70743 |
| B9 | 116.00 | 578 | 0.70745 |
| B9 | 117.00 | 1029 | 0.70746 |
| B9 | 118.00 | 528 | 0.70739 |
| B9 | 119.00 | 208.51 | 0.70789 |
| B9 | 120.00 | 419 | 0.70730 |
| B9 | 121.00 | 642 | 0.70747 |
| B9 | 122.00 | 144 | 0.70744 |
| B9 | 123.00 | 127 | 0.70801 |
| B9 | 124.00 | 44 | 0.70767 |
| B9 | 125.00 | 243 | 0.70763 |
| B9 | 126.00 | 288 | 0.70743 |
| B9 | 127.00 | 193 | 0.70725 |
| B9 | 128.00 | 202 | 0.70721 |
| B9 | 129.00 | 690 | 0.70755 |
| B9 | 130.00 | 359 | 0.70747 |
| B14 | 131.00 | 455 | 0.70755 |
| B14 | 132.00 | 434 | 0.70750 |
| B14 | 133.00 | 508 | 0.70749 |
| B14 | 134.00 | 471 | 0.70746 |
| B14 | 135.00 | 362 | 0.70747 |
| B14 | 136.00 | 163 | 0.70731 |
| B14 | 137.00 | 163 | 0.70731 |
| B14 | 138.00 | 266 | 0.70735 |
| B14 | 139.00 | 75 | 0.72859 |
| B14 | 140.00 | 224 | 0.71131 |
| B14 | 141.00 | 235 | 0.70761 |
| B14 | 142.00 | 234 | 0.72629 |
| B14 | 143.00 | 71 | 0.70721 |

## 4. Results

### 4.1. Field Observation

The base of the Barsarin formation is characterized by decimetre-thick beds (<30 cm) of stromatolites, evaporites (Figure 2B,C) and a prominent cherty level, alternating with well-bedded dolomitic limestones and shaly limestones (Figure 2A). The top of the formation shows massive limestones and dolomitic limestones (Figure 3). The thickness of the formation is 8 m in the studied Barsarin section type.

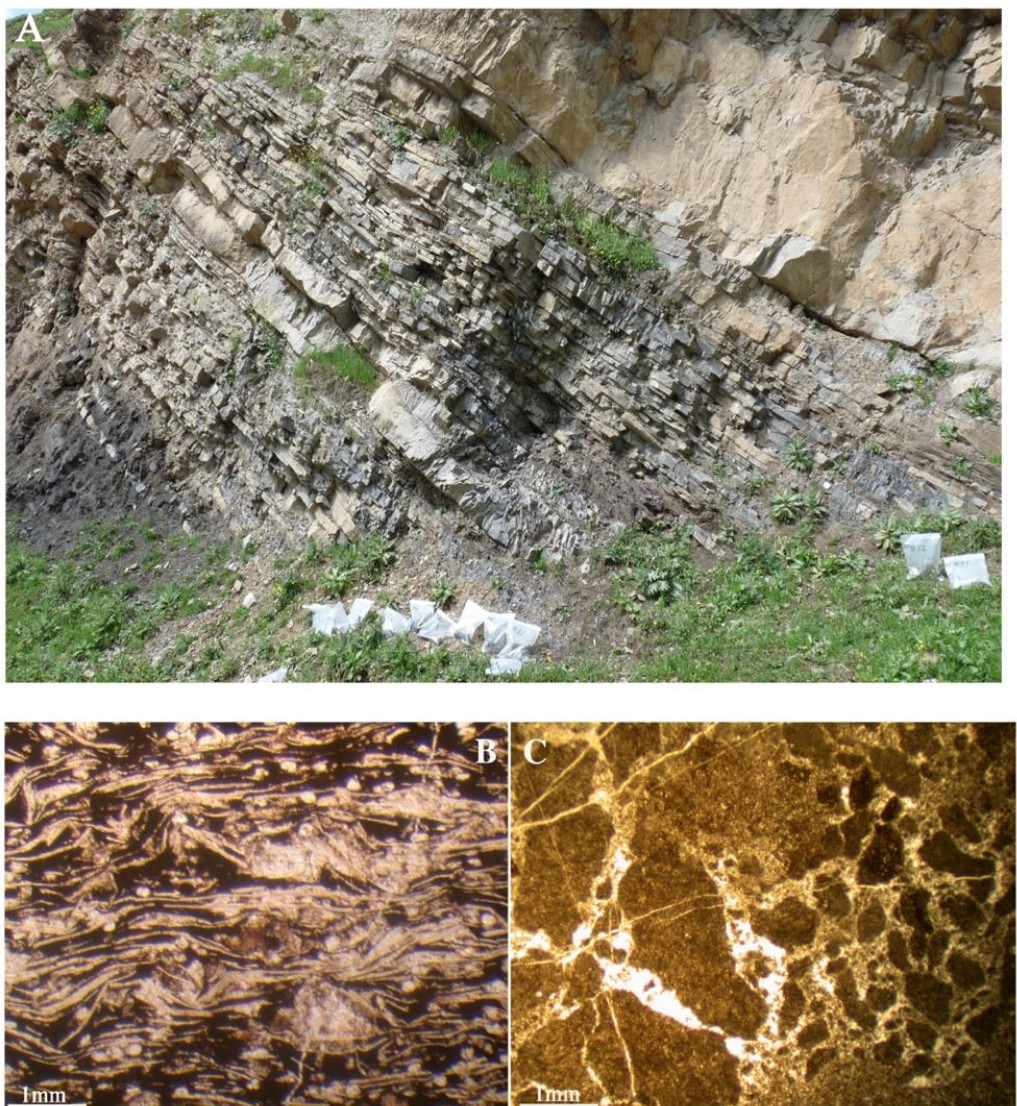

**Figure 2.** (**A**) Photograph showing the exposed section of the well-bedded limestone deposited during Upper Jurassic, Barsarin formation. (**B**) Photomicrograph illustrates the micro-laminated features of Bositra-like pelecypod shells (see [15]), the micro-laminated features cross-cut by vein of evaporites; (**C**) collapsed mudstone microfacies, giving auto-brecciated sediment. The pore spaces of the breccia are completely occluded by evaporates, with no traces of late diagenetic cements.

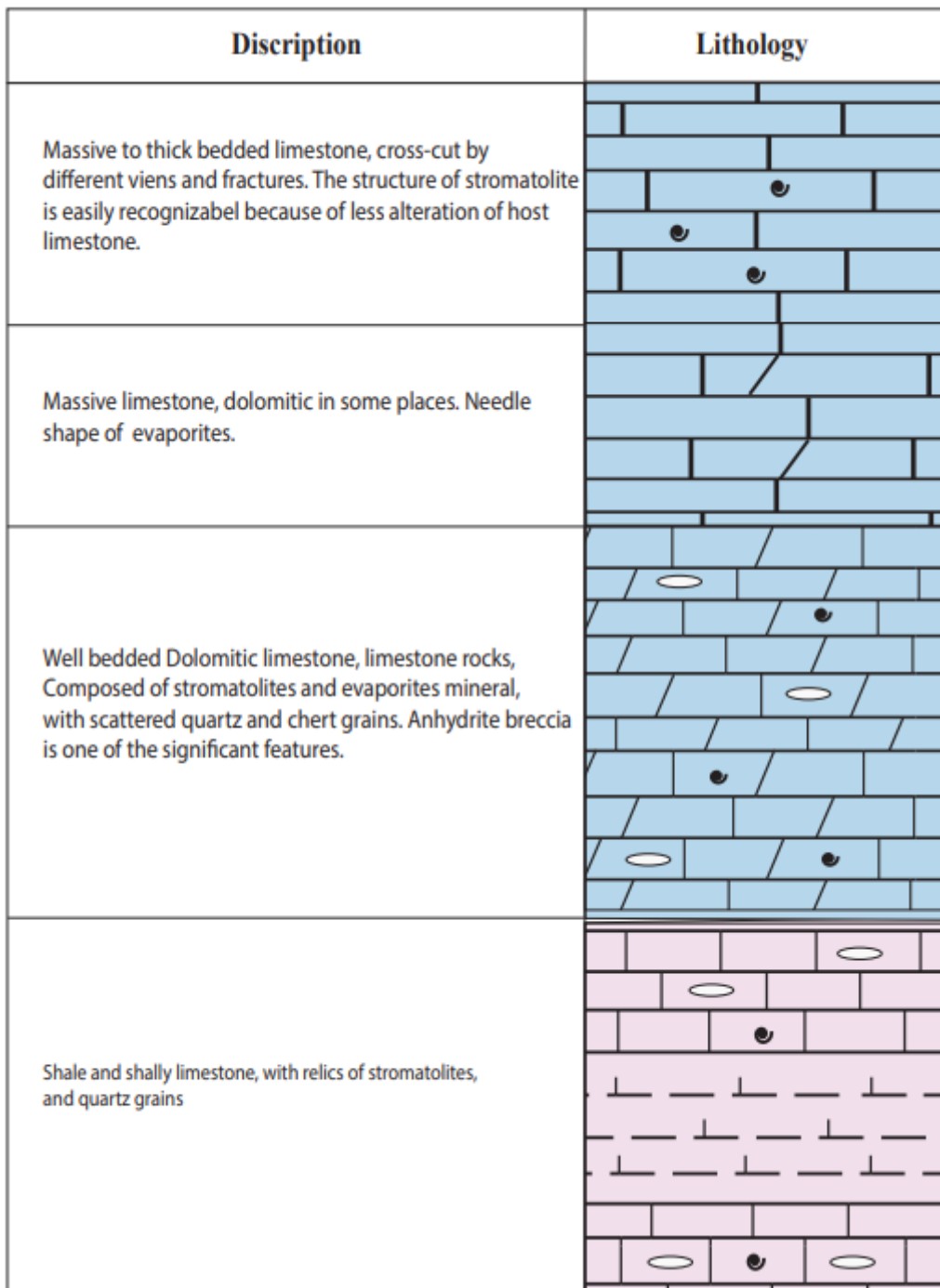

| Discription | Lithology |
|---|---|
| Massive to thick bedded limestone, cross-cut by different viens and fractures. The structure of stromatolite is easily recognizabel because of less alteration of host limestone. | |
| Massive limestone, dolomitic in some places. Needle shape of evaporites. | |
| Well bedded Dolomitic limestone, limestone rocks, Composed of stromatolites and evaporites mineral, with scattered quartz and chert grains. Anhydrite breccia is one of the significant features. | |
| Shale and shally limestone, with relics of stromatolites, and quartz grains | |

**Figure 3.** Master log showing the general lithology of the Barsarin Formation (Not to scale).

The fracture- and void-filling dolomite and calcite are distributed throughout the Barsarin profile, in addition to several vertical calcite filling joints and few horizontal ones in the upper part.

*4.2. Petrography*

The host carbonates show three types of microfacies: (i) mudstones, (ii) radiolarian wackestones and (iii) stromatolite boundstones (Figures 3 and 4). Despite these facies having undergone dolomitization, they are still recognizable, particularly the fine Bositra-like pelecypod shells and stromatolitic fabrics in the lower and parts of the section (Figure 2B).

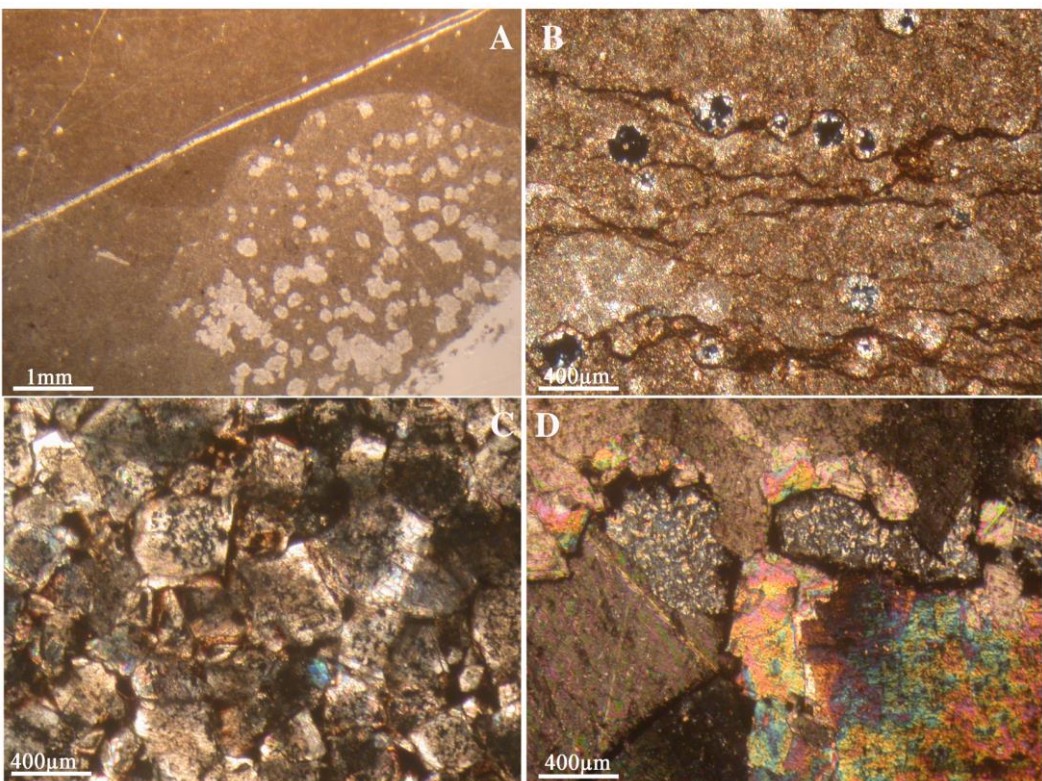

**Figure 4.** Photomicrographs illustrating characteristics of silicate mineral content. (**A**) Litho-clast "right most part of the photo" embedded by mega-quartz grains. (**B**) Micro-organisms of spherical shapes of Radiolaria, the chalcedony micro-crystalline quartz partially filled the pore spaces of Radiolaria. (**C**) Mega-quartz grains in places replaced by dolomite minerals, see the strong relief and inclusions of quartz in the core of some grains. (**D**) Late diagenesis dolomite cements "Dь" replaced the microcrystalline quartz.

Abundant needles/lath crystals from evaporite minerals are found within the mudstone microfacies, which shows an in-situ brecciated fabric with irregular cracks and veins (Figure 2C). Fracture-filling evaporites are volumetrically less important than the needle shaped evaporites (Figure 2A). The in-situ collapse of the pristine facies leading to a breccia is related to the dissolution of the evaporitic minerals (Figure 2C). Sometimes, these breccias are still filled with evaporite minerals.

Dolomite appears as a cement in two forms. $D_m$ represents the initial precipitate, and it is characterized by dark coloured crystals of variable sizes and anhedral-euhedral shapes (Figure 5B,C). $D_m$ shows, in places, a typical rhombohedral shape, with micritic-rich core and transparent cortex (Figure 5B). Larger transparent euhedral crystals are also observed. The second cement occurs in the fractures and vugs and consists of a dirty coarse euhedral dolomite ($D_b$) with curved or non-planar surface. It can also be euhedral with–a characteristic zonation and shows a sweeping extinction under crossed nicols (Figure 5C,D). $D_b$ only occurs in fractures and open spaces. Fracture-filling calcite crystals are always associated with the second dolomite ($D_b$); it displays variable sizes, is transparent and consists of blocky, twinned crystals. Geometric relationships under petrographic analysis show that the blocky calcites were replaced by coarse dolomite ($D_b$). In addition, the detrital grains, mainly quartz, are observed within the host carbonate. They show the traces of dissolution and erosion, and this could be due to the distance of grain transportation (Figure 6A,D).

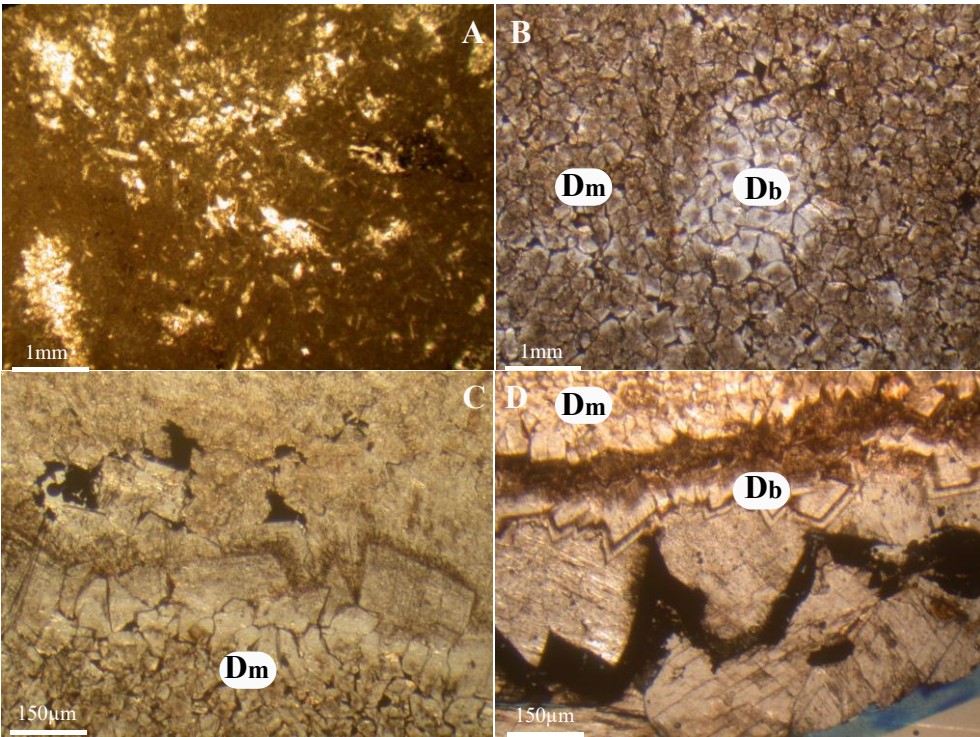

**Figure 5.** Photomicrographs illustrate: (**A**) The needle shape of evaporites within the mudstone microfacies that filled the pore spaces, (**B**) early dolomitizing cement ($D_m$) were mostly preserved the previous pristine facies "dark brown colour", while in the central part of the photo the rhombohedral shape of dolomite which precipitated later. The lower part (**C**) and upper part (**D**) of the photos show the predating cement formation "$D_m$" and the coarse grains of dolomite "$D_b$" from the late dolomitizing fluids.

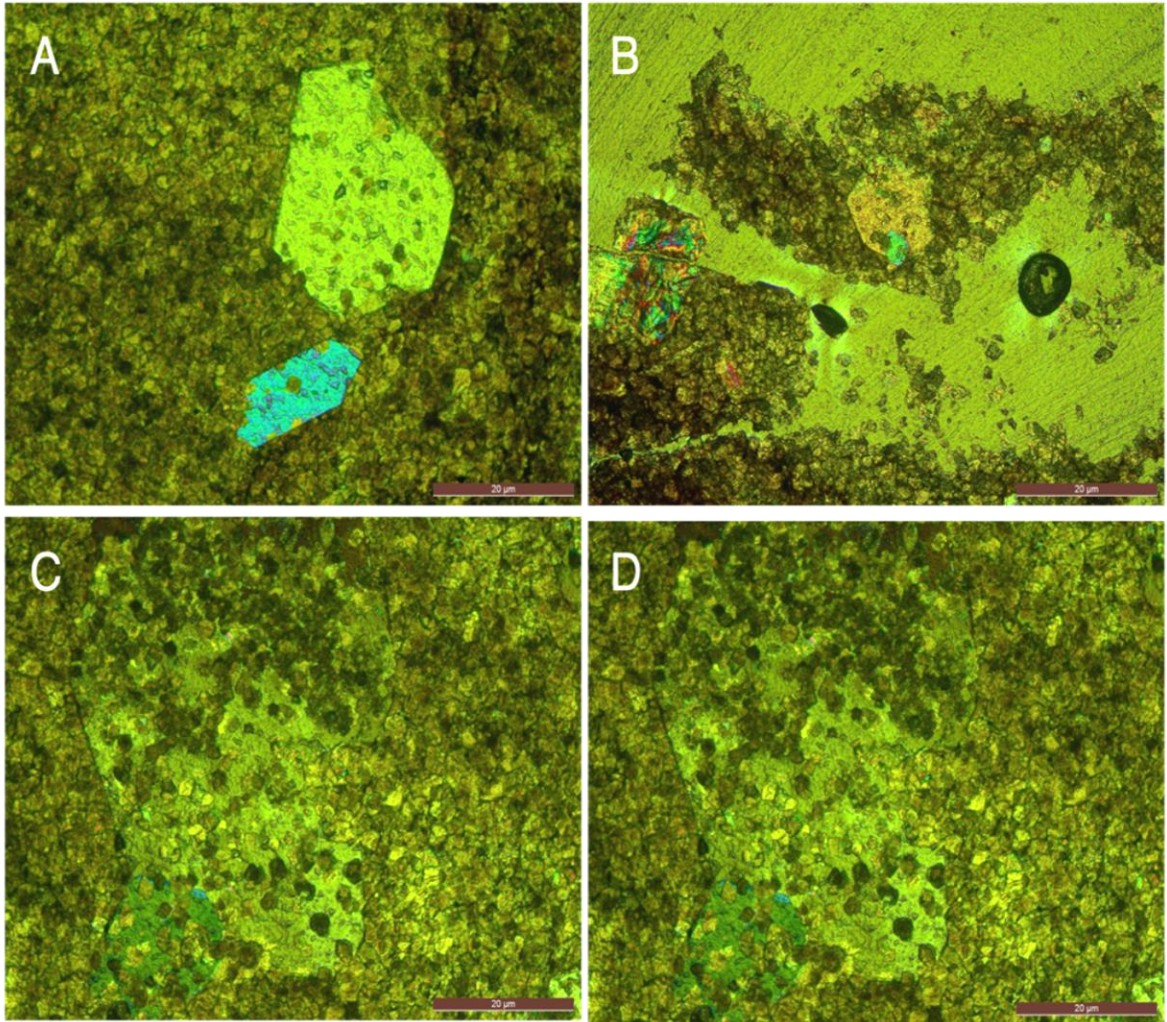

**Figure 6.** (**A–D**) Photomicrographs illustrate the weathered clasts of quartz from outside of the Barsarin basin, close up the eroded outline of polygonal grains (**B**) and other detrital grains in (**D,C**) that embedded in host carbonate rocks.

SEM analyses reveal that the host limestone, dolomite and calcite cements are commonly associated with quartz grains, more specifically at the bottom and upper parts of the formation (Figure 7). The grain size of quartz ranges from micro-quartz to mega-quartz, and in places silica appears in its chalcedony form.

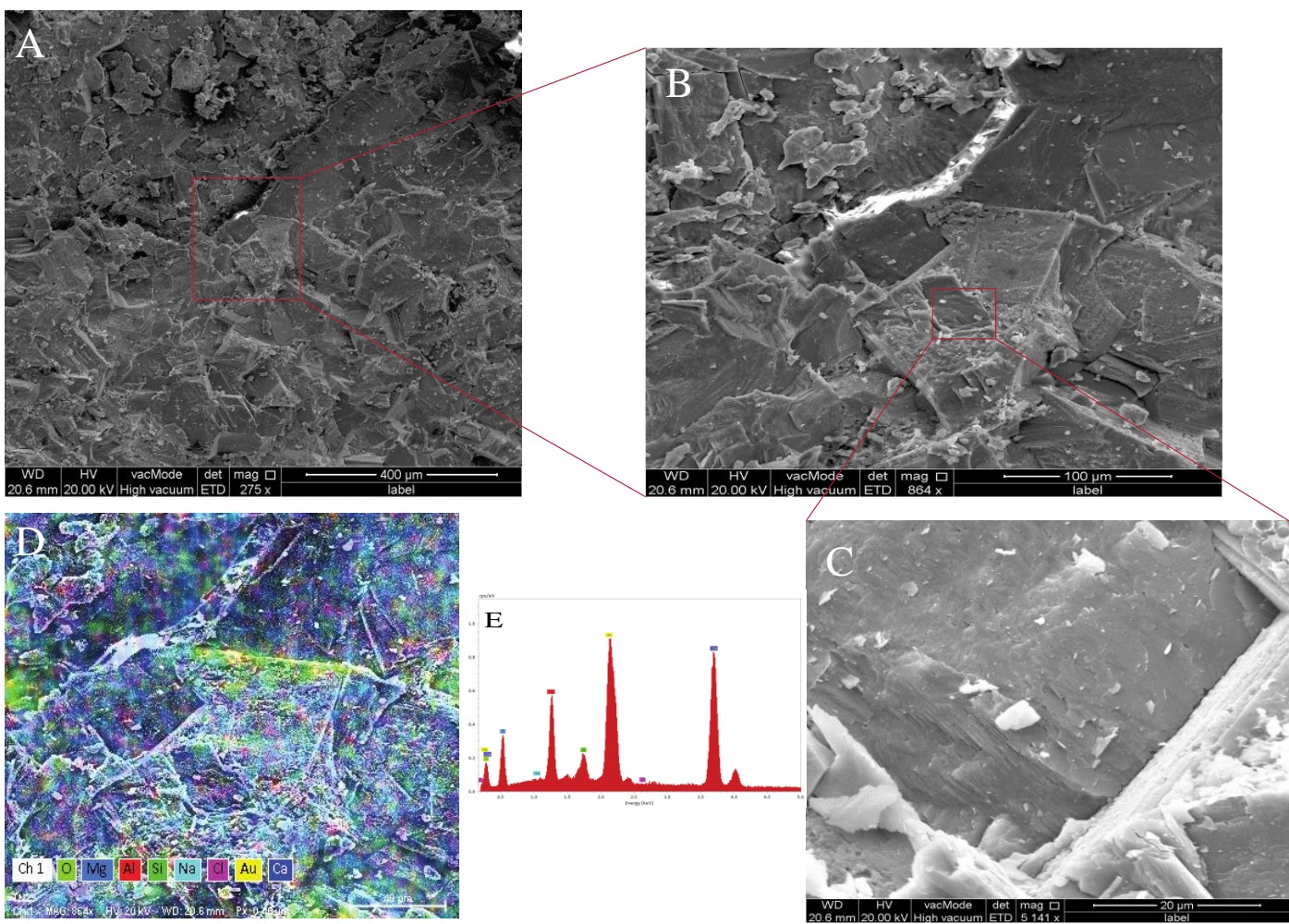

**Figure 7.** SEM/EDX images of the evaporite carbonate rock sample obtained from freshly gold coated surface, the images illustrate: (**A**) the carbonate grains cemented by evaporitic minerals, the circle marks the aggregate grains of gypsum crystals, the rectangle marks two phases of minerals and this part, enlarged in (**B**,**C**), where the central part, composed of euhedral, eroded the surfaces of calcite crystals where cemented by evaporites. (**D**,**E**) Mineralogical mapping of the same sample location (**C**); (**D**,**E**) the majority of sodium element was distributed around the euhedral crystals of calcite.

### 4.3. Oxygen, Carbon and In Situ Strontium Isotopes

The O–C isotope compositions of the pristine facies, calcite, dolomite and evaporite are listed in Table 1. The 23 samples from fracture-filling dolomite and calcite cements, matrix dolomite ($D_m$) and pristine facies of the formation have oxygen–carbon isotopic values populated into two groups. Group I is isotopically heavier in composition than group II (Figure 8). The host limestone and evaporite oxygen–carbon values fall into group I, while the calcite and dolomite cement oxygen-carbon values mostly fall into group II.

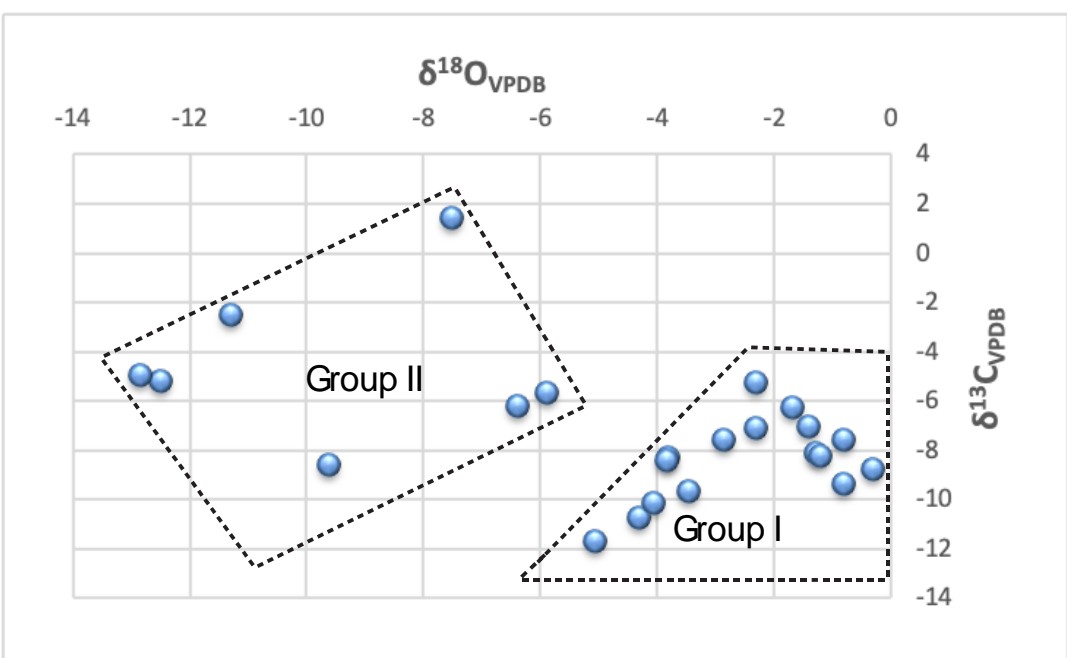

**Figure 8.** Oxygen and carbon isotope compositions in the Barsarin formation (n = 24). The values are populated on two groups: the first, group I (evaporitic mudstones and wackestones), has heavier values than the second, group II (calcite and dolomite cements).

Most of the carbon and oxygen marine isotopic compositions of the Barsarin facies are significantly lower than those of seawater limestone during Late Jurassic time reported by Brand and Anderson [16,17]. Isotope compositions of Late Jurassic belemnites led these authors to define the isotopic composition of seawater during this time, with $\delta^{18}O_{VPDB}$ values ranging from 0.04‰ to −0.99‰ and $\delta^{13}C_{VPDB}$ values from 0.97‰ to 1.51‰. In group I of the Barsarin samples, $\delta^{18}O_{VPDB}$ values are between −5.09‰ to −0.34‰, and $\delta^{13}C_{VPDB}$ between −5.23‰ to −11.66‰. In group II, $\delta^{18}O_{VPDB}$ values are −12.87‰ to −5.90‰, and $\delta^{13}C_{VPDB}$ values are −8.55‰ to +1.47‰. However, a few host limestones that are free of evaporites have lighter oxygen isotopic values (up to −4.34‰) than those containing evaporites (up to −2.33‰, see the in Table 1).

Two samples (B9, B14, Table 2) have been analysed by laser ablation to obtain an absolute high-resolution radiogenic $^{87}Sr/^{86}Sr$ ratios on fracture-filling cements and the carbonate matrix. In-situ measurement of $^{87}Sr/^{86}Sr$ signatures in calcite and dolomite are provided by laser ablation to obtain the absolute radiogenic composition of many measurements on a micrometre-sized scale (MSS) within a single crystal (Figures 9–11). Laser spots analyses by ICP-MS measured several precise points on each crystal in two samples (B9, B14, n = 35). The plotted $^{87}Sr/^{86}Sr$ ratios display two positive excursions (Figures 10 and 11). The first excursion is characterized by the highest radiogenic signatures, ranging from 0.70789 to 0.72859. Dolomite and calcite crystals fall into the second positive excursion, with strontium isotope ratios varying from 0.70721 to 0.70772. They are less radiogenic than the host limestone values. Most of the laser spot measurements from the host limestone show a significant Sr radiogenic ratio, and mostly fall into the first positive excursions. The absolute strontium isotope data of the host limestone is radiogenically very high compared to the data from literature [9], and do not fit with the late Jurassic seawater strontium isotope curve, which varied between 0.70671 and 0.70713.

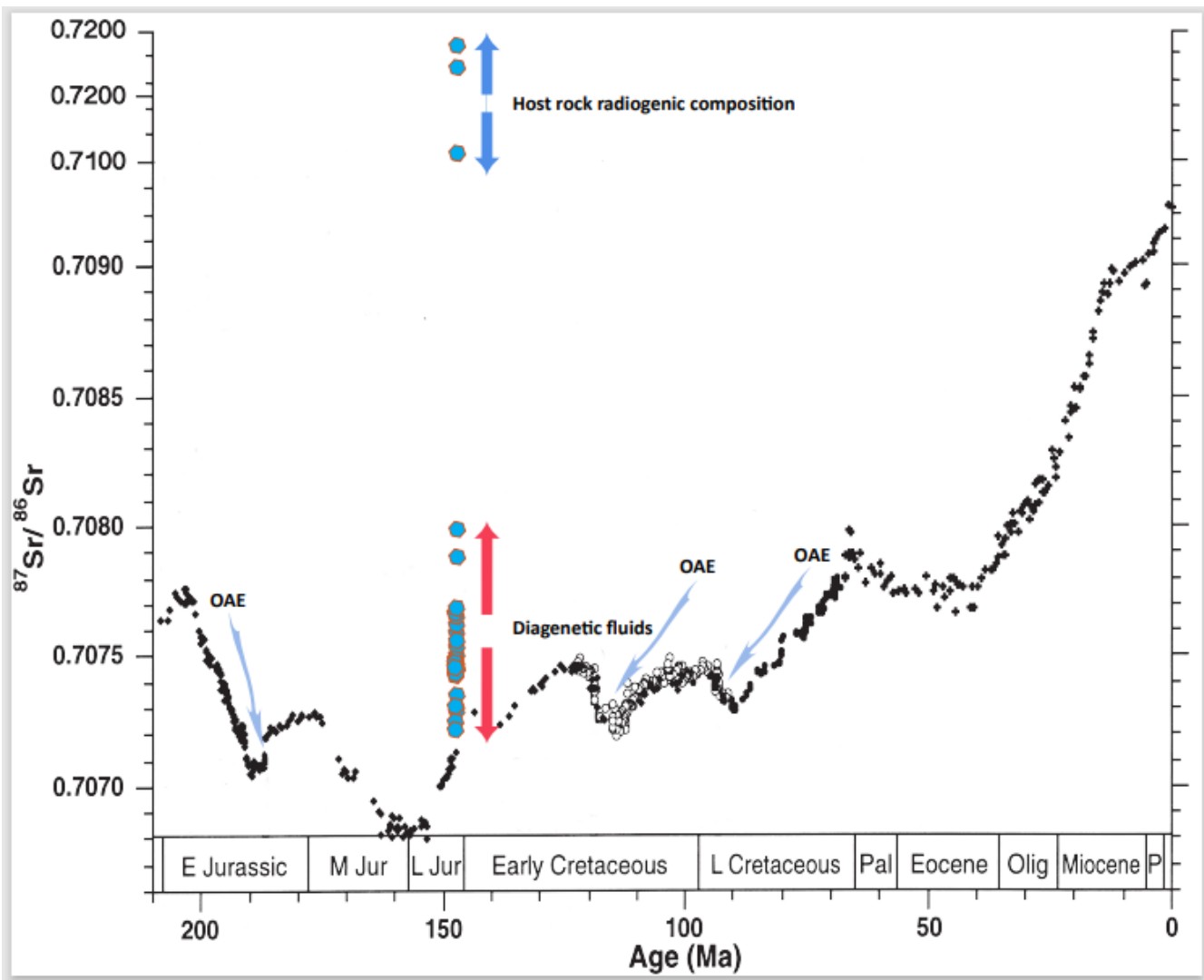

**Figure 9.** The seawater Sr-isotope curve shows minima in the Late Triassic, Pliensbachian-Toarcian, Callovian-Oxfordian, Aptian-Albian, and Cenomanian-Santonian. The recent study data "blue plots" from our study are shown for comparison. The recent data "this study" populated around two positive excursions: the first host carbonate positive excursion, and the second diagenetic carbonate positive excursion. The host carbonate value is highly radiogenic, higher than the whole seawater Sr isotope curve, while the value of carbonate diagenesis samples is still high and less radiogenic than host carbonate samples (modified after Jones and Jenkyns [9]). OAE = Oceanic Anoxic Event.

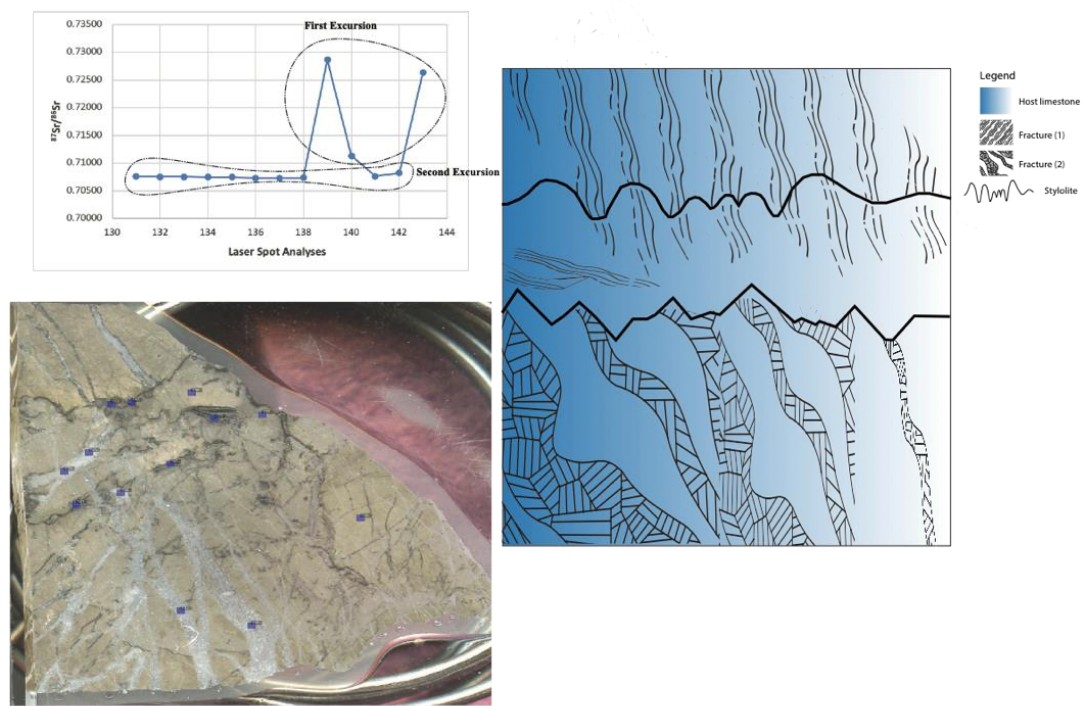

**Figure 10.** The absolute radiogenic strontium isotope values v. the location of the spot analysis "sample No. B14". The location of analysis is represented by cubic blue colour where the ablated spot size used is 213 μm and depth of crater is ~20 μm.

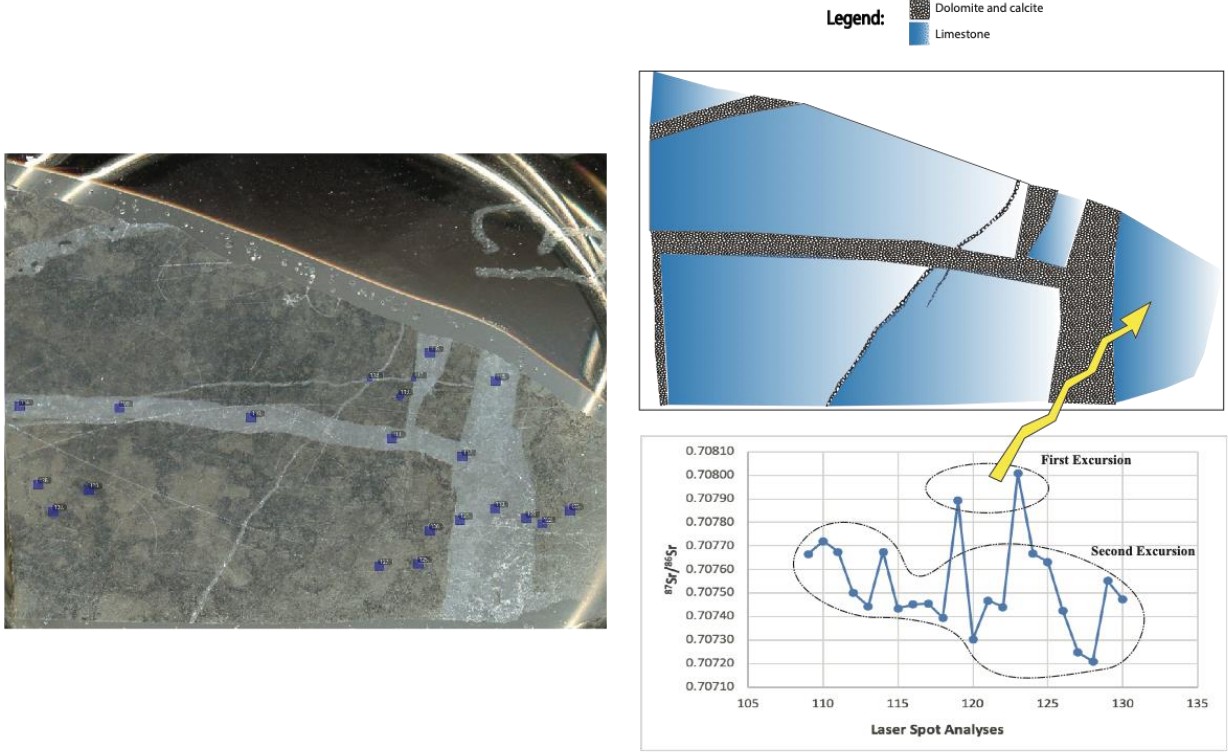

**Figure 11.** The absolute radiogenic strontium isotope values v. the location of the spot analysis "Sample No. B9". The location of analysis is represented by cubic blue color where the ablated spot size used is 213 μm and depth of crater is ~20 μm.

## 5. Interpretation and Discussion of Conventional and Non-Conventional Isotopic Signature Trends

### 5.1. δ13. C$_{VPDB}$-δ$^{18}$O$_{VPDB}$ Isotopes

The estimation of isotope values of marine and diagenetic sediments is useful in providing a better understanding of seawater compositional changes during deposition or post-deposition due to any chemical or biological alteration. The oxygen isotope reconstruction in Mesozoic marine sedimentary rocks shows a general decrease of δ$^{18}$O$_{VPDB}$ values of the global seawater throughout the Late Jurassic [16–18]. Brand and Anderson proposed a past seawater δ$^{18}$O$_{VPDB}$ from around +2.5‰ to −3‰ during Late Jurassic. Furthermore, the general trends of a warming through the Late Jurassic yields a paleo-temperature value in the range of 20–30 °C [19]. δ$^{18}$O$_{VPDB}$ values between −5.09‰ and −0.34‰ of our host limestone (Group I; Table 1) do not fit within the previously documented oxygen isotope values of marine carbonate during Late Jurassic and is probably related to paleo temperature conditions.

Paragenetically, our host carbonate rocks from the Upper Jurassic have significant negative values of oxygen and carbon isotopic compositions, as low as −5.09‰ and −11.66‰, respectively. The host carbonates predated evaporite formation (see Figure 7 and Group I from Figure 8). Host carbonate samples that contain relicts of evaporites have δ$^{13}$C$_{VPDB}$ δ$^{18}$O$_{VPDB}$ values (from 0.34‰ to −2.33‰ and −6.25‰ to −9.30‰, respectively) lower than those of the previous reported marine domain for the same period [20], but higher than those of the pure host carbonates in our study. This trend of wide variation in oxygen and carbon isotope compositions from the host carbonates might be due to fluctuations of sea level, possibly related to climate change, at a local, regional scale (e.g., [21]), if not at a global scale [9], and/or could be linked to a secondary imprint on marine rock composition.

Floating angular clasts of host carbonates are cemented by evaporite crystals, suggesting the evaporative episode (mainly gypsum and halite) post-dated the host carbonates (Figures 2C and 7). This led to the formation of collapse breccias from the host carbonates, probably related to sea-level fall and climate change, as measured previously by the ratio of precipitation to evaporation (e.g., [22]). Hence, the large depletion in oxygen and carbon values, together with systematic occurrence of evaporates, suggest that dry and warm climate conditions drove the primary oxygen signal to negative values, where oxygen isotopic composition of seawater in the study basin increased during precipitation (e.g., [23]).

Evaporite minerals reflect the hypersaline evolution of the basin that led to heavier oxygen isotopic values than those of marine limestone (e.g., [24]) [25]in contrast to the carbon isotopic composition [26]. Nevertheless, our $^{18}$O depleted values in Group I are lighter than those of the oxygen isotopic composition of the marine curve. δ$^{18}$O$_{VPDB}$ depleted values in host carbonates (Group 1) reveal that the former evaporative episode was affected by climatic changes during Late Jurassic, possibly through river input during a humid season [26]. Therefore, the depleted oxygen values from our study, combined with published data for the Late Jurassic-Early Cretaceous interval, suggest that global and gradual warmer palaeotemperatures occurred during this interval when the Barsarin formation was deposited. A similar scenario has previously been reported by Lécuyer and Allemand [27] and Vickers[28], from clumped isotopes and conventional oxygen isotope data, to interpretate the climatic change throughout Jurassic Cretaceous times. Furthermore, the covariance between oxygen and carbon isotopic compositions in marine carbonates (Figure 8) is probably linked to period boundaries with climate change [29,30].

The O–C isotopic values of second group (Group II) are lighter in their oxygen and heavier in their carbon isotope composition than those in Group I, although some of the carbon values overlap the marine signature. Group II has distinct and wide variations of oxygen–carbon isotopic compositions, compared to Group I. Group II is influenced by late diagenetic fluids that had a role in precipitation of calcite and dolomite cements (Figure

5). The $\delta^{18}O_{VPDB}$ values of diagenetic cements exhibit more depleted values than those not influenced by diagenetic fluid alteration (Figure 8). This could be due to the long-term interaction between the host rock and the diagenetic fluids, promoted by fluids heated up to 80 °C or more (e.g., [31]). Therefore, the considerable and significant variation of $\delta^{18}O$–$\delta^{13}C$ from Group II could be linked to the evolution of fluids (e.g.,[31]), with slow precipitation rates under subsurface conditions (e.g; [32]), or rapid precipitation of dolomite and calcite due to abrupt change in pressure and temperature [8,33]. The extremely negative $\delta^{18}O$ values in the coarse-sized dolomite and calcite crystals (Figure 8) highlight the interaction of the host limestones with diagenetic fluids (e.g.,[8]) as indicated in Group II. Since the oxygen isotope values are extremely negative in deep diagenetic conditions, where the mechanism for dolomite and calcite precipitation is slow, hot circulation is the most probable source for samples belonging to Group II. Nevertheless, Group II is populated in two $\delta^{18}O$ assemblages. One follows the J trend of meteoric water with extreme depleted $\delta^{13}C$ values, and the second shows lighter oxygen isotope compositions with slight changes in $\delta^{13}C$ values. These are quite consistent with the mixing of hot and meteoric fluids.

The carbon isotope values in Group II also exhibit a wide variability in their composition, although some values still possess host rock signatures (Group I: Figure 8). This could be related to the buffered system during deposition of diagenetic cements (e.g., dolomite and calcite) where the diagenetic fluid mixed with rock-derived carbon of the previous carbonates (Group I).

Considering the quartz grains in the host carbonate, the lower and upper parts of Barsarin formation are richer in silica than those of the middle part. Since the oxygen isotope composition of carbonates rich in silica is lighter than seawater, the chemistry of water facies rich in silica is not related to a marine source. Therefore, the origin of the quartz grains is not related to diagenetic evolution, as previously reported in a similar case by Bustillo [34], despite the outer rims of the chert or quartz having been often replaced by dolomite/calcite in a late diagenetic event (Figure 4D). This case has been reported also by Bustillo [35].

### 5.2. Sources of Variation of Absolute Strontium Isotopes from the Late Jurassic and Onwards, Utilizing Non-Conventional Laser Ablation ICP-MS

Seawater $^{87}Sr/^{86}Sr$ ratios varied homogeneously through geological time, thus making this isotope proxy an important chemostratigraphic tool [36,37]. The evolution of the seawater $^{87}Sr/^{86}Sr$ ratio curve during the Jurassic-Cretaceous is widely driven by variation in continental weathering rates (riverine) and predominant oceanic fluxes of non-radiogenic strontium sporadically during the Jurassic ]5[. Silica hosted carbonates are another source that impacted the seawater radiogenic composition curve [38]. These sources of radiogenic strontium isotopes had a different effect on the seawater strontium isotope record (Figure 8, [9,5]). Our data provides, for the first time ever, a high-resolution strontium isotope composition from pristine facies and associated diagenetic carbonates using non-conventional direct laser ablation (Figures 10 and 11) in Upper Jurassic carbonate source rock.

The diagenetic imprint has to be precisely considered since the diagenetic fluids are impacted by the uptake or loss of Sr radiogenic isotope and Sr concentration. Therefore, the petrographical characteristics and paragenetic sequence for the entire section is critical before interpreting the strontium isotopic analysis. In the Barsarin formation early silicification and evaporite minerals formation is possibly syn-depositional [39] or occurred during a very early diagenetic event (e.g., [35]). The collapse and brecciation of the mudstone facies suggest that the host sediments and evaporite minerals were coincident with the depositional time of Barsarin formation. Silica infilling the intragranular porosity is also observed within dolomite and calcite lattice and suggests a late diagenesis process. The co-occurrence of silica and diagenetic carbonates bring the

idea of silica recycling and/or continuous flux "input" of silica during and after the depositional setting.

The $^{87}Sr/^{86}Sr$ ratios are populated into two positive excursions. The first excursion gives a radiogenic isotope value of up to 0.72859 in the host carbonate rocks; these values are three times higher than those reported during Late Jurassic times [9]. The second excursion observed in the calcite and dolomite cements is less radiogenic than host carbonates. The core of carbonate cement shows impurities due to growth of quartz grains in their crystal framework (Figure 4C). These two positive excursions of absolute strontium radiogenic data could be explained in the following two ways:

(i) The silicate sediments that precipitated from seawater were converted into fibrous quartz and/or micro- (i.e., the biogenic silica-rich sediments) and mega-quartz via dissolution–reprecipitation processes [40]. The first excursion in pristine facies, that often contains silicate minerals in the studied samples, has lower Sr concentration and higher radiogenic strontium isotope values (high silicate minerals). The silicates were observed through the whole thickness of the carbonate facies, and as well as solid inclusions within the core of the calcite and dolomite crystal lattices. These inclusions could be the reason for the enrichment of the strontium ratio in the carbonate cements. Therefore, silica cycling via dissolution–reprecipitation processes could be one possible scenario for the observed radiogenic strontium, mainly for the first positive excursion (Figures 10 and 11). The silicate sediments in host carbonate show no corrosion or disturbed grains due to dissolution of the siliceous sediments or replacement, but in places the groundmasses are embedded with silicate grains totally different from the host carbonate facies, as highlighted frequently by their different colours from those of the facies (Figure 4). This indicates that some of the silicate sediments likely derive from outside of the Barsarin depositional basin, i.e., they are siliceous sediments continental detritus (Scenario II).

(ii) The second positive excursion relates to the diagenetic cement, where the strontium isotope ratio and strontium contents formed two assemblage groups: the highest strontium ratios with lowest strontium concentration (first group) and the lowest strontium ratios with highest strontium contents (second group). In addition, the weathered and eroded particles embedded in host carbonate (Figure 6A–D) reveal that this phase it could be derived from the mixing of detrital silicate minerals transported by rivers from the catchment "low Sr content" with high diagenetic Sr content "hot fluids".

The strontium isotopic composition of seawater is predominately controlled by inputs at mid-ocean ridges, through hydrothermal exchange ($^{87}Sr/^{86}Sr$ = ~0.703), and from rivers, through continental erosion ($^{87}Sr/^{86}Sr$ = 0.705 to > 0.800; mean ~0.712; Palmer and Edmond, 1989). Variations in seawater ($^{87}Sr/^{86}Sr$ reflect changes in the concentration of Sr and the $^{87}Sr/^{86}Sr$ of dominant sources, [18]), with the hydrothermal source being less radiogenic (~0.703), generally account for less of the total Sr input in the ocean system [41]. The concentration and isotopic value of the riverine flux is highly variable due to heterogeneities in the isotopic composition of continental crust, the concentration of Sr in source rocks and weathering rates.

The increased input of hydrothermal strontium at oceanic ridges had a major effect on seawater strontium isotope composition during the Jurassic [11]. Hydrothermal sources decreased seawater $^{87}Sr/^{86}Sr$ ratios sporadically during the Bajocian–Callovian period, while, during mid-Bajocian, elevated submarine volcanism increased the $^{87}Sr/^{86}Sr$ ratios (cf. [36]; Figure 9). The enhanced hydrothermal strontium input culminated with the global sea-level rise at the Middle-Late Cretaceous [42]. However, more published data showing a degree of scatter confirm the uniform rise of seawater strontium isotopic curve from Late Jurassic–Early Cretaceous times [23]. During the Late Jurassic, major events, like early Mesozoic rifting and Jurassic subduction, have been suggested to occur during the Zagros Orogeny [43]. Stratigraphic and geochronological keys suggest that a large volume of detrital continental crust contaminated by mantle-derived magmas impacted Late Jurassic times [44]. These authors concluded that the strontium enrichment was either derived from an enriched mantle source or acquired by crustal assimilation.

Although, by the Late Jurassic the highest $^{87}Sr/^{86}Sr$ ratio reported by Lechmann [44] was as large as 0.70697, which is very low compared to our study (values reaching 0.72859).

The widespread values along one host limestone sample's "first excursion" has a higher radiogenic isotope ratio, of as much as 0.72859 (Figure 10), although the ratio in the seawater reference curve did not exceed 0.70720 during Late Jurassic times (Figure 9). Davies and Smith [45] documented that the strontium isotope ratio increased relative to pristine values by releasing strontium through the interaction of hydrothermal "hot" fluids with siliciclastic basement, or by release from continental siliciclastics [9]. The latter source is considered as radiogenic when evolved from riverine input [46]. Linking the continental siliciclastic source with our petrographical observations, the host carbonate rocks of the Barsarin formation show that silicification came from radiolarian cherts (Figure 4B). Yet, Radiolaria are not sufficiently abundant to be a sink for strontium isotopes. The early diagenetic siliceous cement is related to the source of silica from Radiolaria [47], and the primary intraparticle porosity was filled by chalcedony chert, as occasionally observed in our study area. Different-sized lithoclasts are embedded in the host carbonate rocks and contain variable sized-silicate grains (Figure 4A). These embedded silicate grains exclude that the radiogenic strontium was provided from the radiolarian cherts. Hence, at least the first radiogenic excursion was controlled by transport-limited nature of the weathering reaction, the same case was reported by Palmer and Edmond [41].

In SE France, hydrothermal dolomite, hosted in Upper-Jurassic sediments, were discovered with similar radiogenic values (ratios reaching 0.71120, Salih et al., 2020b). In this case the dolomite was related to radiogenic sources derived from riverine input. Furthermore, uplift in the latest Jurassic initiated detrital inputs related to the erosion along the Zagros Mountains [43]. Thus, the first positive excursion of an absolute strontium ratio would indicate an increase in weathering rates due to flux of radiogenic strontium.

The second positive excursion from the diagenetic cement samples evolved from diagenetic fluid sources. The dolomite and calcite cements preserved the inclusions and traces of silicate materials (Figure 4C). The remanent of solid inclusions inside the dolomite and calcite thereby limits the strontium isotope ratio. The remanent of silicate minerals either originated from host carbonates during replacement process or from interaction of hot fluids with radiogenic basement [5]. The origin of dolomite from hot fluids would preferentially record the interaction of hot fluids interaction with a radiogenic basement (cf., [5]), or the radiogenic source together with negative $\delta^{13}C$–$\delta^{18}O$ linked to seepage of low-temperature meteoric water in subsurface carbonate rocks [48].

### 5.3. Origin of $\delta^{13}C_{VPDB}$ $\delta^{18}O_{VPDB}$-Light and $^{87}Sr/^{86}Sr$-Rich Sediments

The enrichment and depletion of strontium and oxygen–carbon isotopes in carbonates is mainly controlled by sea-level fluctuation, the erosion of continental sediment and the recycling of ocean water through mid-oceanic ridges [49,50]. The absolute $^{87}Sr/^{86}Sr$ ratios of the host rocks and diagenetic cements in Barsarin formation show two positive radiogenic excursions, which are higher than those reported in the Upper-Jurassic marine carbonates [9]. The two positive excursions that have a linear extrapolation with Sr concentration, suggesting that $^{87}Sr/^{86}Sr$ ratios could be released from meteoric ($^{87}Sr/^{86}Sr$ ratios of up to 0.72859 and Sr concentration of 75 ppm) in the host carbonate rocks (Figure 12).

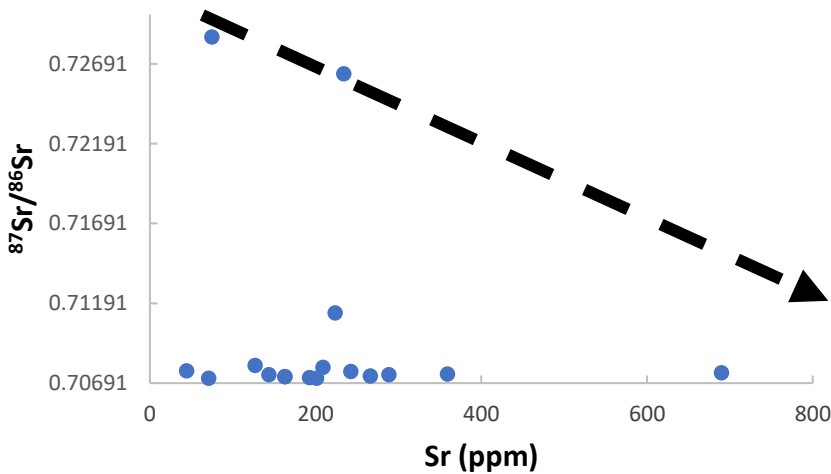

**Figure 12.** Strontium concentration v. [87]Sr/[86]Sr ratios in host carbonate samples; note the variable data of Sr concentration and wide range of Sr isotope ratios.

The deviation of our Sr isotopes values from the Sr isotope-seawater curve could be linked to sea-level fall during Late Jurassic, with a relatively warmer climate in the Kimmeridgian [51]. Sea-level fall is further supported in our study by the co-occurrence of evaporitic minerals floating inside the mudstone and in-situ brecciated mudstones. These observations are consistent with a first positive excursion of the strontium isotope ratio (up to 0.72859) and the heavier oxygen-isotope composition in those samples containing evaporites compared to those of host carbonates free of evaporites. These arguments suggest the beginning of sea-level fall and dry climatic conditions during the deposition of the Barsarin formation in Late-Jurassic times.

This discussion can be further evaluated by oxygen and carbon isotopes and petrographic examination: if we considered that high Sr-isotope ratio and low Sr concentrations represent fresh water mixed with evaporated seawater in the host limestones, then $\delta^{13}C$ and $\delta^{18}O$ values would be depleted in their composition. This is the case in Group I, where oxygen-isotope composition shows a wide range, being heavier in the host limestones lacking evaporite. This wide range in Group I suggests the involvement of continental input by a fresh-water river source during the deposition of the Barsarin formation. As a consequence, $\delta^{13}C$ and $\delta^{18}O$ values from the host limestones that closely match the first positive excursion of strontium isotopes could be linked to a short-lived arid episode during Late Jurassic. A global warm climate will increase the rate of chemical weathering [18]. Therefore, warmer climates must increase both riverine Sr flux and its [87]Sr/[86]Sr ratio [9]. This is well-supported by the dissolution and brecciation of the mudstones cemented by evaporites (Figure 2C).

The nature of the fluids involved in and after deposition of any carbonate is influenced by the composition of precipitated carbonates, therefore, other environmental conditions need to be considered. The late carbonate cements that contain silica showed relatively high radiogenic signatures (Figures 10 and 11) matching closely the radiogenic isotope values of hot fluids that interacted with the basement or with bedrocks rich in silicate minerals [5,45]. The Sr-isotope values of the dolomite/calcite cements fell into the second positive excursion of [87]Sr/[86]Sr values (up to 0.70772) and could be linked in the Barsarin formation to the interaction of hot circulation fluids with the basement recrystallized rocks or interaction of hot fluids with evaporites or siliciclastic rocks (e.g., [52,53]). The negative values of oxygen isotope compositions from dolomite and calcite cements suggest a hot diagenetic fluid involvement [33]. However, the calcite

precipitation from meteoric water could also be associated with an enrichment of strontium isotopes and the depletion of $\delta^{13}C$ and $\delta^{18}O$ values, but the negative covariant trend of $\delta^{13}C$ and $\delta^{18}O$ values exclude the involvement of meteoric water [30]. Furthermore, the lower limit of marine isotope concentration is about 200 ppm [54], while the strontium concentration from diagenetic carbonate is populated in two assemblage groups between 148–195 ppm and 362–1029 ppm (Figure 13). The lowest Sr content, with highest strontium isotopes, indicate a meteoric-derived water, while the highest strontium content with lowest strontium isotopes indicates a possible influence of hot fluid in the diagenetic system. The co-occurrence of dolomite and calcite crystals better supports our scenario, in which dolomite and calcite are attributed to mixing of two different fluids. Therefore, the first positive excursion of direct $^{87}Sr/^{86}Sr$ ratio by laser ablation from the host limestones is partly linked to the weathering riverine strontium input during the depositional time of the Barsarin formation along the ZTFB. The second positive excursion of $^{87}Sr/^{86}Sr$ ratios is most likely related to post-depositional processes, involving two mixed sources of diagenetic fluids in the subsurface setting.

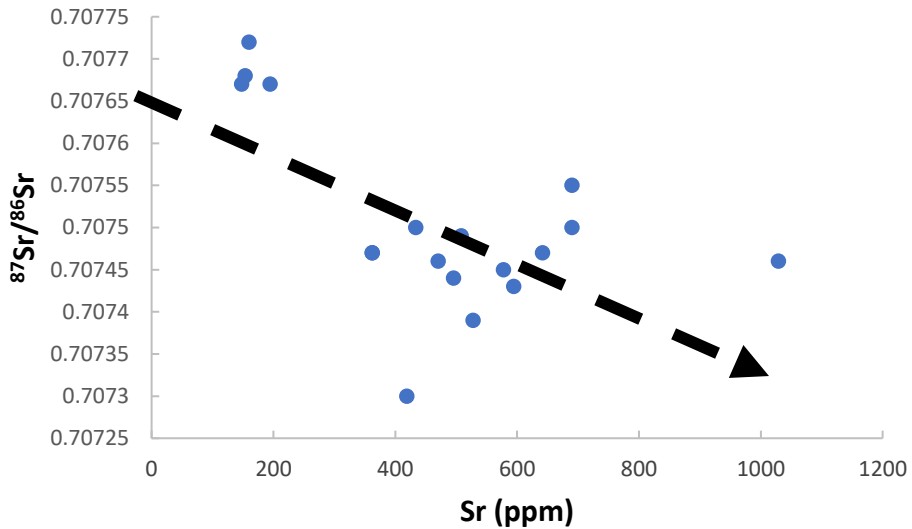

**Figure 13.** Strontium concentration v. $^{87}Sr/^{86}Sr$ ratios in diagenetic samples, the high strontium concentration is consistent with wide range of $^{87}Sr/^{86}Sr$ ratios, and the low strontium concentration is consistent with high $^{87}Sr/^{86}Sr$ ratios.

## 6. Conclusions

1. The origin and evolution of host carbonates and the diagenetic fluid history from Barsarin formation were identified on the basis of fieldwork, petrography and geochemistry ($\delta^{13}C$–$\delta^{18}O$ and high resolution $^{87}Sr/^{86}Sr$ isotope ratios by laser ablation on a micrometer-sized scale -MSS).

2. The Barsarin carbonate sequence contains silica infilling the intragranular porosity. Silica is also very abundant, as are inclusions in the dolomite and calcite cements that filled the fractures and pore spaces that affected the formation. The presence of silica indicates that simultaneous diagenetic processes affected the host rocks and cement formation.

3. $\delta^{13}C$–$\delta^{18}O$ isotope values were populated into two groups. The host carbonates of Group I had isotopically heavy oxygen compositions compared to the diagenetic cements of Group II. Group II showed two different assemblages of oxygen-carbon isotope values, suggesting that mixed diagenetic fluids were involved in the precipitation of the carbonate cements.

4. Two positive excursions of the strontium isotope-ratio curve have been identified. The first excursion is related the host carbonates, and the second is the fracture-filling carbonate cements.

5. Two scenarios were proposed for the first positive excursion of the strontium ratios. The first suggests that a silica cycling occurred through dissolution-reprecipitation processes, while the second scenario linked the high values of strontium ratio to weathering of detrital silicate rocks in the headwaters of a fresh river mixed with evaporated seawater. The carbon and oxygen isotopes, and the two assemblage groups of the first excursion of strontium isotopes and strontium contents exclude the first scenario and indicate that the host carbonates formed in a marine evaporative environment mixed with fresh water originated from riverine-rich silicate detrital "Second scenario".

6. The closely associated silica contents with the first positive excursion of absolute $^{87}Sr/^{86}Sr$ ratio in the host carbonates are related to the continuous flux of riverine input during Late Jurassic.

7. Silica and evaporites with low Sr concentration, high $^{87}Sr/^{86}Sr$ ratios and wide range of oxygen and carbon isotope compositions point to a riverine weathering input in the evaporated seawater during the sea level fall which occurred during the Scenario II event.

8. The wide range of $\delta^{13}C$–$\delta^{18}O$ values in Group II and their positive covariance suggest a late fluid mixing during the second excursion of the $^{87}Sr/^{86}Sr$ ratio. In this case mixed fluids, consisting probably of low-temperature meteoric and hot diagenetic fluids, were inferred from the stable isotope compositions of the post-depositional carbonates "calcite and dolomite cements".

**Author Contributions:** writing and preparation the original draft, N.S. and A.P.; review and editing, K.K., J.N P., N.S and A.P., Methodology and software, A.G, A.P. and N.S. All authors have read and agreed to the published version of the manuscript.

**Funding:** The study benefited from research funds of the Université Libre de Bruxelles (ULB)-Belgium.

**Acknowledgments:** This work was supported by Université Libre de Bruxelles-Belgium and University of Alberta-Canada.

**Conflicts of Interest:** The authors declare no conflict of interest.

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
