# Peer review of "Tracking the Origin and Evolution of Diagenetic Fluids of Upper Jurassic Carbonate Rocks in the Zagros Thrust Fold Belt, NE-Iraq"

_water, doi:10.3390/w13223284_

Round 1

Reviewer 1 Report

Dear authors,

as you know, silica diagenesis is an important factor on releasing water in the system you describe.. You may sevarious papers on the subject, one of them is presented below

https://onlinelibrary.wiley.com/doi/10.1111/bre.12168

You manuscript needs rewriting , taking in consideration the role of the diagenesis of the biogenic cilica which is hosted in the sedimentary rocks you studied.. Some note are presented as sticky notes on the text

Author Response

Answer to 1st Reviewer`s comments and suggestions:

First of all, I would like to express my grateful to the 1st Reviewer for his strong contribution in revising the manuscript in details, and below kindly find the answer to all his/her comments and suggestions:

  1. In Abstract, I rewrite the abstract according to your comments in lines: 26 and 28, and I removed the Secondary fluids in line 33.

  1. In introduction part you refer to the chalcedony that could be diagenetically derived from biogenic silica - opal-A [diatoms, radiolaria, sponge spiqules, sillicoflagellate skeletons? And you proposed some sources for example: https://onlinelibrary.wiley.com/doi/10.1111/bre.12168, and other relevant works.

Answer to 1st Reviewer: Different sources of silica-rich sedimentary rocks have been added and using the following references; (Maliva and Siever, 1989; Wrona et al., 2015; Gao and Land, 1991; and Salih et al., 2020).

  1. at the bottom left, put the geographical map of Iran and the location of the studied area.

Answer to Reviewer: The geographic map has been added accordingly.

  1. you may introduce a figure showing the studied outcrop(s);

Answer to Reviewer: Please see Figure 1A

  1. omit this sentence

Answer to Reviewer: The sentence has been removed and the sentence later was also modified accordingly.

  1. Move Figure 2A in the geological setting you describe earlier

Answer to Reviewer: The figure is moved accordingly

  1. Nothing to see in figure 3, you have to solve this problem.

Answer to Reviewer: this problem is solved, however in my file all is fine, but I will ask the editorial board to double check it again..

  1. how you identify that the needles are evaporite minerals, which of them?

Answer to Reviewer: The Gypsum is most likely observed under optical microscope (please see Figure 3A) Barsarin, beside the compositional isotopic value is support this event..

  1. you describe here silica diagenesis from an original opal-A to chalcedony.. Note that the radiolarian cells are more resistant in dissolution than the diatom frustule and so they remain mostly at their original shape.. it means we have solid-solid inversion of opal-A to opal-CT and finally to chalcedonic quartz..

Answer to Reviewer: Yes, your comment is right, but do you think I should go to details about Opal A to Opal-CT since I focused on origin of radiogenic composition of host rock, which is not from the host silica if this is the case, I added also another plate to ensure your answer (please see figure 6).

  1. Figure 8, poor quality

Answer to Reviewer: the same technical problem like (point 7), I solved this problem.

  1. you have to define them, halides, sulphates, carbonates?

Answer to Reviewer: the floated grains is carbonate while the cemented materials are mainly gypsum and halite (all these added in text).

  1. You may have detrital and authigenic quartz.. you have already detected chalcedonic silica related with radiolarian cells..

Answer to Reviewer: yes I have recognized the chalcedonic texture rich silica, but also I have a lithoclast from outside of this paleoenvironment (please see figure 4 A,C, D), beside the radiogenic isotope values is out of Jurassic marine carbonate and so much higher.

  1. the biogenic silica-rich sediments..

Answer to Reviewer: the sentence is added accordingly.

  1. I think as I understood I have to add silica minerals

Answer to Reviewer: the sentence is added accordingly

  1. you have to prove it, i.e absence of biogenic silica relics.. SEM measurements are needed..

Answer to Reviewer: the high radiogenic strontium isotope which is recoded precisely by LA-IC-PMS on most spot point of Upper Jurassic sample `s` (sometimes up to 0.72859) is support that either the system was in close contact with enriched mantle source or acquired by crustal input. The mantle components did not recognized, while the silicate minerals is usually found where also contained continuously Al, Si, Na, Cl, as major elements.

I added also another plate (figure 6), which explain that the detrital clasts coming from outside of the studied basin.

  1. Replacement process?

Answer to Reviewer: I mean the replacement is one of the diagenetic processes

  1. silica minerals. what are the silicate minerals you suggest?

Answer to Reviewer: Mainly quartz

  1. poor quality (Figure 11)

Answer to Reviewer: this is technical problem, from my file is ok, it will be solved when you receive it.

  1. Reference section was modified and corrected according to Reviewer comments.

Thanks again for your valuable comments, which really make a manuscript in a better form.

Reviewer 2 Report

BRIEF SUMMARY

I present my corrections/remarks below and if  authors improve/answer, I could give the “green light” for publication of this work in “Water” journal.

SPECIFIC COMMENTS

  1. Line 16: Don’t have in bold the word “utilizing”..
  2. Lines 18-20: Avoid in the whole text the words like “we”, as it sounds selfish. Replace “we constrain” with “are constrained” and write this in the end of the sentence, i.e., “The origin and evolution of the…..petrography and fieldwork are constrained”.
  3. Line 89: Why do you write “Fig. 1A”? Do you mean “Fig. 1”?
  4. Table 1, Line 119: Each Table and Figure must be included (with their titles/explanations) in the same page and not to be splited in two pages. Here, begin Table from next page 4.
  5. Line 278: Begin paragraph 5 in the next page 14, in order the title and the subtitle to be in the same page with the main text.

6 Line 386:  Replace “we present” with “are presented” and write this in the end of the sentence, i.e., “The Sr/Sr ratios…. two positive excursions are presented”. Similarly, in the whole text avoid words like “we” and “us”.

7.References, Line 576: Re-control all the references in order to check that the details of each one (names of authors, titles, No. of pages and volumes) are correct. You should also reassure that all the references are appeared inside the text.

I believe that if the authors follow my suggestions, and

answer  ALL my questions,

this paper could be suitable for “Water” Journal.

I would like to check it one more time before the final publication

Author Response

Answer to 2nd Reviewer`s comments and suggestions:

Dear Reviewer, I am grateful for your contribution to add your comments in this project in order to be in a better shape for publication, Kindly find my answer and you will find the modification inside the text too (Word file):

  1. Line 16: Don’t have in bold the word “utilizing”..

Answer to Reviewer: The bold word is mofified accordingly.

  1. Lines 18-20: Avoid in the whole text the words like “we”, as it sounds selfish. Replace “we constrain” with “are constrained” and write this in the end of the sentence, i.e., “The origin and evolution of the…..petrography and fieldwork are constrained”.

Answer to Reviewer: the word we is removed and has been modified accordingly.

  1. Line 89: Why do you write “Fig. 1A”? Do you mean “Fig. 1”?

Answer to Reviewer: you are absolutely right, I removed the letter A.

  1. Table 1, Line 119: Each Table and Figure must be included (with their titles/explanations) in the same page and not to be splited in two pages. Here, begin Table from next page 4.

Answer to Reviewer: I think it is technical problem, in word file is ok, but for sure I will contact the editorial office to check it again..Thanks

  1. Line 278: Begin paragraph 5 in the next page 14, in order the title and the subtitle to be in the same page with the main text.

Answer to Reviewer: Now they are in the same page

6 Line 386:  Replace “we present” with “are presented” and write this in the end of the sentence, i.e., “The Sr/Sr ratios…. two positive excursions are presented”. Similarly, in the whole text avoid words like “we” and “us”.

Answer to Reviewer: I modified the text accordingly.

7.References, Line 576: Re-control all the references in order to check that the details of each one (names of authors, titles, No. of pages and volumes) are correct. You should also reassure that all the references are appeared inside the text.

Answer to Reviewer: I double check every reference that cited with their detailed information.

Thanks again for your valuable comments, which really make a manuscript in a better form.
